# BUILDING RELIABLE LONG-FORM GENERATION VIA STEP-WISE HALLUCINATION REJECTION SAMPLING

## ABSTRACT

Large language models (LLMs) have achieved remarkable progress in open-ended text generation, yet they remain prone to hallucinating incorrect or unsupported content, which undermines their reliability. This issue is exacerbated in long-form generation due to hallucination snowballing, a phenomenon where early errors propagate and compound into subsequent outputs. To address this challenge, we propose a novel inference-time scaling framework, named Step-wise HAllucination Rejection Sampling (SHARS), that allocates additional computation during decoding to detect and reject hallucinated content as it is produced. By retaining only confident information and building subsequent generations upon it, the framework mitigates hallucination accumulation and enhances factual consistency. To instantiate this framework, we further introduce a new uncertainty-based hallucination detection method, named HalluSE, for long-form generation, improving upon the prior semantic entropy approach. The combined system enables models to self-correct hallucinations without requiring external resources such as web search or knowledge bases, while remaining compatible with them for future extensions. Empirical evaluations on standardized hallucination benchmarks demonstrate that our method substantially reduces hallucinations in long-form generation while preserving or even improving the informativeness of generation.

## 1 INTRODUCTION

Large language models (LLMs) (OpenAI, 2025; Yang et al., 2025a; Grattafiori et al., 2024) have markedly expanded the frontiers of artificial intelligence, demonstrating impressive capabilities in open-ended text generation across domains such as question answering (Min et al., 2023; Wei et al., 2024), code synthesis (Jimenez et al.), and scientific communication (Lu et al., 2024). However, their practical deployment is hindered by a persistent and well-documented challenge: hallucination (Ji et al., 2023). Hallucinations arise when models generate content that is factually inaccurate, unsupported, or in conflict with the provided input (Bang et al., 2025), often delivered with high fluency and confidence. This phenomenon undermines the reliability of model output and user trust, and poses risks in high-stakes applications.

Hallucinations are particularly concerning in open-ended generation, where the extended and unconstrained nature of the outputs makes it especially challenging to validate. In addition, prior studies (Zhang et al., 2024; Zhao et al., 2025; Yang et al., 2025b) have shown that longer generations tend to amplify hallucination risk, a phenomenon known as *hallucination snowballing*, in which early errors propagate and trigger additional mistakes. This underscores the importance of intervening early in the generation process to interrupt error accumulation and thereby reduce hallucinations.

Separately, a growing body of research (Wei et al., 2022; Yao et al., 2023; Muennighoff et al., 2025; DeepSeek-AI et al., 2025) has investigated the paradigm of inference-time compute scaling, which improves model performance by allocating additional computation at generation time. This paradigm is particularly well-suited for hallucination mitigation in high-stakes domains such as healthcare, scientific discovery, and law, where users are often willing to accept slower responses in exchange for more factual and reliable outputs. Nevertheless, this direction remains underexplored, and to the best of our knowledge, there are no well-established findings on how inference-time scaling affects factuality in open-ended generation.

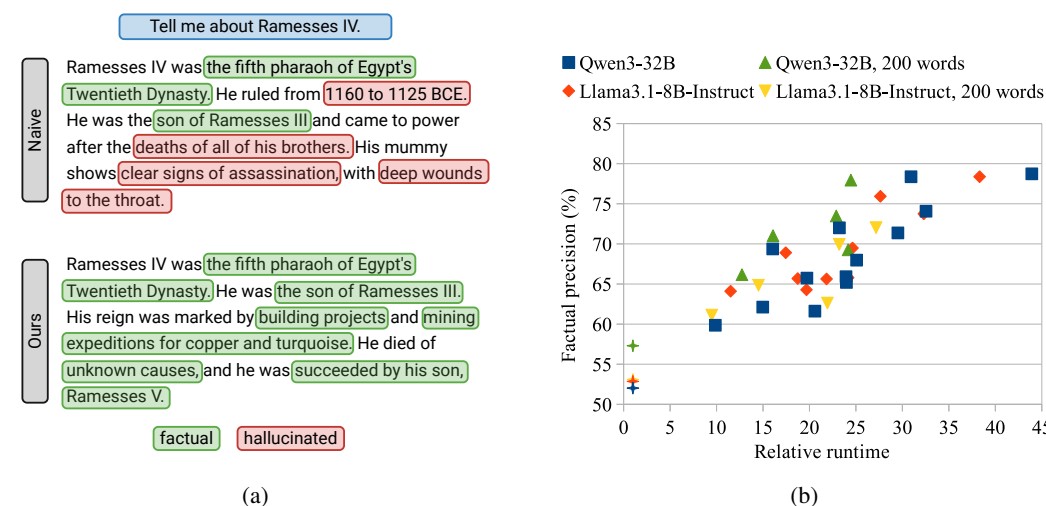

(a)             (b)

Figure 1: **(a) Comparison of biographies generated by Greedy decoding and our method**. Unlike Greedy decoding, our method rejects hallucinated content, preserves factual information, and acquires additional factual content (the last two sentences in the displayed generation) beyond the original information space. **(b) Scaling of factual precision with respect to inference-time computation on the FactScore benchmark** (Min et al., 2023). Inference-time computation is approximated by relative runtime, measured as a factor of the runtime of the corresponding Greedy decoding method for each setup. Each data point in the figure represents an individual run of our method under one of the four setups, except for the leftmost star point of each color indicating the Greedy decoding baseline. Full experimental details are provided in Section 5.1.

Inspired by these insights, **we introduce a general inference-time compute framework, termed Step-wise HAllucination Rejection Sampling (SHARS), to mitigate hallucinations in open-ended generation**. SHARS leverages an arbitrary detector to identify and reject hallucinated content as it is produced during generation, preserves only factual segments, and builds subsequent outputs upon them (Fig. 1a). This design aims to increase the proportion of factual information in the final output while disrupting hallucination snowballing from its early stages. To instantiate this framework, **we further propose a new uncertainty-based hallucination detection method, HalluSE, tailored for long-form generation**. HalluSE builds upon the prior semantic entropy approach (Farquhar et al., 2024), incorporating several refinements to address its limitations and improve detection effectiveness. Notably, SHARS is designed to be detector-agnostic, allowing it to integrate with any hallucination detection method and thereby broadly benefit from future advances in hallucination detection research.

We conduct extensive experiments on diverse long-form factuality benchmarks, including FactualBio (Farquhar et al., 2024), FactScore (Min et al., 2023), and LongFact (Wei et al., 2024), to evaluate our methods. Empirical results show that HalluSE significantly improves hallucination detection accuracy over prior approaches in long-form generation. SHARS further proves effective in mitigating hallucinations in open-ended generation while preserving, and in some cases enhancing, output informativeness. Importantly, **SHARS exhibits a promising scaling property: when appropriately configured, factuality continues to improve as additional inference-time computation is allocated within a certain range** (Fig. 1b). For instance, SHARS improves factual precision by about 26% for evaluated models on the FactScore benchmark.

## 2 RELATED WORKS

**Hallucination detection**. Farquhar et al. (2024) introduced hallucination detection via semantic entropy, which estimates uncertainty in the space of meanings by clustering diverse model samples and measuring entropy over the induced semantics. They benchmarked this method against two alternatives: Self-Check, where the model verifies its own assertions, and P(True), which measures

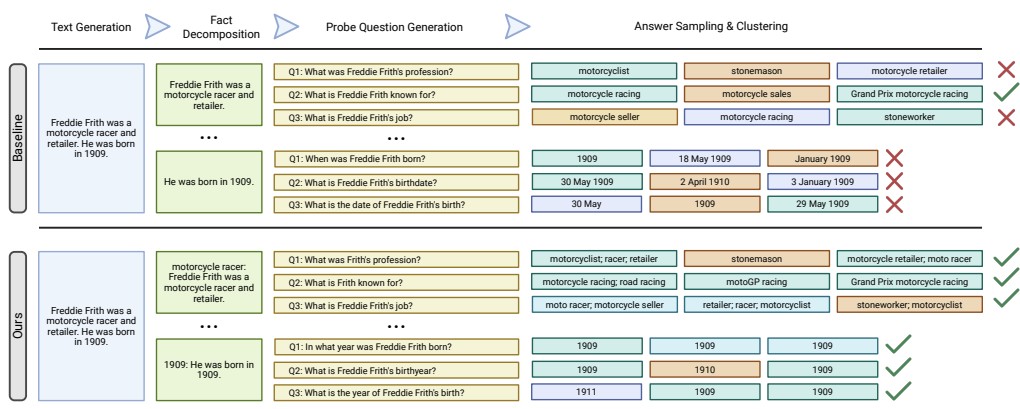

Figure 2: Illustration of naive long-form semantic entropy method and our proposed HalluSE. Different colors under 'Answer Sampling & Response' denote distinct semantic clusters of generated responses. A green check indicates low semantic entropy (high agreement, reliable answers), while a red cross marks high semantic uncertainty (likely hallucinated content).

the probability that the model predicts the token "True" when few-shot prompted to compare a main answer with alternatives. Another training- and retrieval-free approach by Mündler et al. (2024) detects hallucinations by eliciting multiple responses and identifying contradictions or inconsistencies. Other methods train lightweight probes. For instance, Kossen et al. (2024) trained probes to approximate semantic entropy from hidden states of a single generation, while Obeso et al. (2025) trained probes on web-search-grounded, entity-level labels to detect hallucinations in real time. Alternatively, Min et al. (2023), Wei et al. (2024), and Zhao et al. (2025) detect hallucinations by decomposing generated text into atomic facts and checking them against trusted external sources.

**Hallucination mitigation**. To mitigate hallucinations, Tian et al. (2024) fine-tune models using preference data generated from a retrieval-enabled judge and direct preference optimization (DPO), enabling the model to prefer factual responses. Huang & Chen (2024) propose FactAlign, which assigns sentence-level factuality rewards to reinforce supported spans in long-form outputs. Gu et al. (2025) introduce Mask-DPO, which masks non-factual sentences during preference optimization so that updates focus exclusively on factual content.

Inference-time mitigation approaches have also been explored. Integrative decoding (Cheng et al., 2025b) aggregates self-consistent continuations by jointly selecting supported tokens. Chuang et al. (2024) propose DoLa, which reweights next-token probabilities by contrasting logits from late and early layers. Retrieval-augmented generation (RAG) (Lewis et al., 2020) grounds generation on retrieved passages to replace unsupported spans, while Cai et al. (2024) improve RAG by introducing outline-guided generation and factuality-aware optimization for web-augmented long-form outputs. More recently, Cheng et al. (2025a) incorporate tree search–based algorithms to enable explicit slow-thinking generation, mitigating hallucinations during inference.

# 3 HALLUSE: DETECTING HALLUCINATIONS IN LONG-FORM GENERATION

This section introduces HalluSE, our uncertainty-based hallucination detection method for long-form text generation. HalluSE builds on the prior semantic entropy approach for long-form generation (Farquhar et al., 2024), while addressing several of its key limitations. In Section 4.1, we further employ HalluSE as the hallucination detector to instantiate our primary hallucination mitigation framework.

## 3.1 BACKGROUND: SEMANTIC ENTROPY AND HALLUCINATION DETECTION

Semantic entropy (Farquhar et al., 2024) is an uncertainty measure that captures the variability of a model's predictions in the semantic space rather than the token space. Instead of only considering

surface-level probability distributions over tokens, semantic entropy groups candidate generations into meaning-equivalent clusters and measures the entropy across these clusters. Given a set of candidate generations sampled from the model, each generation is mapped into a semantic cluster $C_i$. The probability of a cluster is defined as the sum of token-level probabilities of all generations assigned to it: $p(C_i) = \sum_{y \in C_i} p(y)$, where $p(y)$ is the model probability of generation $y$. The semantic entropy is then computed as the entropy over cluster probabilities: $H_s = - \sum_i p(C_i) \log p(C_i)$. The full technical details can be found in Farquhar et al. (2024).

Low semantic entropy indicates semantic agreement among candidate generations, while high semantic entropy reflects semantic disagreement and is often associated with hallucinations. This property makes semantic entropy a natural signal for hallucination detection: when the model is confident and semantically consistent, the likelihood of hallucination is lower, whereas high semantic entropy often correlates with unsupported or erroneous content.

## 3.2 NAIVE LONG-FORM SEMANTIC ENTROPY AND ITS LIMITATIONS

The semantic entropy method described above assumes that candidate answers are short-form. To extend it, Farquhar et al. (2024) proposed a naive approach for applying semantic entropy to long-form generation. As shown in Fig. 2, the generation is first decomposed into a set of fact claims. For each fact claim, several probe questions with expected answers are generated to query the fact, and the short-form semantic entropy method is then applied to each question. This procedure effectively reduces long-form hallucination detection to a series of short-form detection tasks.

The naive long-form semantic entropy method faces two main limitations as illustrated in Fig. 2. First, it decomposes a generation into fact claims without distinguishing which entity within each claim should be validated. This ambiguity can cause the wrong entity to be probed downstream. For example, given the query `Tell me about Alan Turing` and the generation `Alan Turing is a computer scientist`, the entity of interest is clearly `computer scientist` rather than `Alan Turing`. However, the prior method may incorrectly generate probe questions such as `Who is a computer scientist?`.

---

**Algorithm 1:** Pseudocode of HalluSE.

**Data:** text, $c$, $M$, $Q$, $A$, $\theta$
**Result:** verified facts, hallucinated facts
verified_facts, hallued_facts ← [ ], [ ]
facts ← decompose_facts($M$, text)
**for** *(entity, claim) in facts* **do**
    questions ← gen_questions($M$, $Q$, entity, claim)
    $H_s$ ← [ ]
    **for** *question in questions* **do**
        $H_s$ ← $H_s$ ∪ semantic_entropy($M$, $A$, $c$, question)
    $H_s$ ← mean($H_s$)
    **if** $H_s < \theta$ **then**
        verified_facts ← verified_facts ∪ (entity, claim)
    **else**
        hallued_facts ← hallued_facts ∪ (entity, claim)

**return** *verified_facts, hallued_facts*

---

Second, the naive approach assumes that each probe question has only a single valid answer, so any uncertainty in sampled answers is attributed solely to the model. In practice, however, probe questions can admit multiple valid answers. For example, in biographies, a prominent individual may hold multiple professions. A probe question such as `What is XX's profession?` may thus have several correct answers. Even if the model consistently samples correct but different professions, the resulting semantic entropy remains high, incorrectly flagging the fact as hallucinated.

## 3.3 HALLUSE

HalluSE addresses the limitations of the naive long-form semantic entropy method through three key refinements as illustrated in Fig. 2. First, it decomposes each generation into pairs of entities and fact claims. Second, it improves the prompting strategy with clearer instructions, structured formatting, and few-shot examples. In particular, HalluSE guides the LLM to generate probe questions with unambiguous expected answers, thereby reducing unnecessary cases of multiple valid answers.

**Algorithm 2:** Pseudocode of SHARS.

**Data:** User query $q$
**Result:** Verified response to the user query
verified_text, hallued_text ← "", ""
**while** *not End_Of_Sequence* **do**
    sent ← next_sent($M$, $q$, verified_text, hallued_text)
    verified_facts, hallued_facts ← detect_hallu($M$,
     sent, verified_text)
    **if** *len(verified_facts) = 0* **then**
       | hallued_text ← hallued_text + sent
    **else**
       **if** *len(hallued_facts) ≠ 0* **then**
         | sent ← rewrite_sent($M$, $q$, verified_facts)
       verified_text ← verified_text + sent
       hallued_text ← ""
    // break if no verified_facts for $N$
      times in a row
**return** *verified_text*

**Algorithm 3:** Pseudocode of next sentence sampling.

**Data:** $q$, verified_text, hallued_text
**Result:** A new sentence
text_sofar ← ""
input ← $q$ + verified_text +
 hallued_text
**while** *True* **do**
    token ← next_token($M$, input)
    text ← decode(token)
    text_sofar ← text_sofar + text
    sents ← split_sents(text_sofar)
    **if** *len(sents) ≥ 2* **then**
       sent ← sents[0]
      | break
    input ← input + text
**return** *sent*

Third, it explicitly instructs the LLM to provide all valid answers, when applicable, in each sampling step. The complete HalluSE pipeline is as follows:

1. **Fact Decomposition**: given a model response, HalluSE decomposes it into a set of facts, where each fact consists of an entity and a claim describing a piece of atomic information about that entity from the model response.

2. **Question Generation**: for each fact, HalluSE generates $Q$ probe questions in which the entity and claim serves as the expected short-form and long-form answer, respectively.

3. **Answer Sampling**: for each probe question, it produces $A$ answers conditioned on the preceding context $c$, i.e., the text appearing before the fact in the response.

4. **Semantic Entropy Computation**: semantic entropy is computed from the sampled answers per question and averaged across the $Q$ questions, yielding the semantic entropy of the fact.

5. **Hallucination Identification**: A fact is classified as hallucinated if its semantic entropy exceeds a predefined threshold $\theta$; otherwise, it is deemed factual.

The full procedure is summarized in Algorithm 1. Fact Decomposition, Question Generation, and Answer Sampling are implemented by prompting a pretrained instruction-following LLM, denoted as $M$, with the specific prompts detailed in Appendix A. Semantic Entropy Computation is implemented with its discrete formulation (Farquhar et al., 2024). The LLMs for Fact Decomposition and Question Generation can be arbitrary, while the LLM for Answer Sampling should match the model used to produce the given response. In this work, we employ the same model for all components, including response generation.

## 4 SHARS: Step-wise HAllucination Rejection Sampling

**Motivation**. We observe that open-ended questions admit an effectively infinite range of relevant information that can constitute a valid answer, yet in practice models draw on only a limited subset of this space when generating responses. Intuitively, if hallucinated content in the initial generation can be filtered out and the model is guided to explore the remaining information space for truthful content to fill these gaps, the resulting generation can be free of hallucinations. Moreover, by dynamically grounding generation on truthful information, this process could potentially disrupt the error compounding caused by earlier mistakes and increase the likelihood of sampling factual content.

## 4.1 STEP-WISE HALLUCINATION REJECTION SAMPLING

Following this motivation, we propose our general inference-time compute framework, SHARS, which leverages an arbitrary detector to identify and reject hallucinated content during generation. SHARS partitions the generation into multiple steps—sentences in our setting—and applies hallucination rejection sampling sequentially as each sentence is produced. For a given sentence, hallucination rejection sampling invokes a hallucination detector to assess its factuality. Based on the detection outcome, the sentence is either (i) discarded if it contains no factual information, (ii) rewritten to remove hallucinated content if it mixes factual and hallucinated information, or (iii) retained if it is entirely factual, with no hallucinations detected. Generation terminates when one of the following occurs: (1) an end-of-sequence (EOS) token is sampled; (2) the maximum new-token budget is reached; or (3) fully hallucinated sentences are sampled in $N$ consecutive attempts. The full procedure is summarized in Algorithm 2.

Our method differs from conventional rejection sampling, also known as the best-of-N sampling, for inference-time scaling in three key aspects. First, rejection sampling is performed in a step-wise and dynamic manner rather than applied once to the entire generation. Second, we sample one candidate sentence at a time and resample only when the current sentence is rejected, instead of generating multiple candidates simultaneously. Third, in cases where a sentence contains both factual and hallucinated information, we rewrite it to remove hallucinations rather than discarding it entirely. The latter two strategies improve efficiency and make the approach more practical for inference-time deployment.

**Hallucination detector**. SHARS is designed to operate with any detector by treating the hallucination detector as a black box. In this work, we instantiate SHARS with our HalluSE detector proposed in Section 3.3, serving as the primary hallucination mitigation approach. We adopt HalluSE because (1) it is domain-agnostic and does not require training a new model, and because (2) it does not rely on external tools or reference knowledge sources. These properties allow seamless integration into SHARS and enable zero-shot application across new domains.

We acknowledge that hallucination detection is an active research area, and many alternative methods (Aichberger et al., 2024; Duan et al., 2024; Manakul et al., 2023; Chen et al., 2024) may be suitable for integration into our framework. We leave this exploration for future work, as our current choice already achieves substantial reductions in hallucinations and significant improvements in factual precision compared with existing state-of-the-art mitigation methods, as shown in Section 5.1.

HalluSE estimates the uncertainty of the knowledge probed by generated questions and uses it as a proxy for the uncertainty of the corresponding fact. For example, consider the fact to be verified: `Alan Turing is an athlete`. The relevant knowledge in this case is Alan Turing's profession. If the model is uncertain about this knowledge, it suggests that not only the underlying fact is likely hallucinated, but also that alternative sampled facts for the profession are likely hallucinated. While this improves the efficiency of hallucination detection, it also raises a challenge for sentence sampling: how do we generate a new sentence with knowledge distinct from that in the hallucination?

**Sentence sampling**. To address the above challenge, we explore two strategies, termed Temperatures and Following. The Temperatures strategy gradually increases the decoding temperature for sampling a new sentence as the number of consecutive hallucinated sentences grows. In other words, the longer the model is stuck at a given point in generation, the more randomness is introduced to encourage exploration of alternative continuations. This approach leverages the model's inherent stochasticity to produce diverse sentences, but it can be less efficient as it does not explicitly incorporate information from the identified hallucinated sentences.

In contrast, the Following strategy temporarily retains the identified hallucinated sentences in the generation and samples the next sentence by continuing the generation process, as illustrated in Algorithm 3. This leverages the model's inherent content planning ability to reduce the likelihood of repeatedly generating content about the same knowledge. For example, a model will typically avoid generating a second birthday for an individual once one has already been stated. However, this approach risks allowing hallucinations to influence subsequent generation. To mitigate this effect, we clear the pool of hallucinated sentences whenever new factual information is identified and retained, preventing the pool from becoming excessively large, as shown in Algorithm 2. Furthermore, hallucinated sentences are used solely for sentence sampling and are not passed to HalluSE as context

Table 1: Performance of the baseline and our methods on the FactScore benchmark without constraints on response length. The best score for each metric is highlighted in bold. The ID results for Qwen3 models are omitted because the authors' released code is incompatible with Qwen3.

| Model | Method | Response Rate (%) | No. Unsupported | No. Supported | Factual Precision (%) |
|---|---|---|---|---|---|
| Qwen3-4B | Greedy | 91.2 | 11.2 | 11.2 | 50.0 |
| | DoLa | **94.5** | 11.1 | 11.9 | 51.8 |
| | ChatProtect | 91.8 | 9.7 | 10.8 | 52.6 |
| | Self-Endorse | 92.3 | 6.6 | 9.2 | 58.2 |
| | Ours-Resp | 92.9 | 8.8 | 14.9 | 63.0 |
| | Ours-Info | 89.0 | 8.2 | 16.3 | 66.6 |
| | Ours-Prec | 69.8 | **5.7** | **16.2** | **74.0** |
| Llama3.1-8B | Greedy | **99.5** | 5.7 | **6.7** | 53.7 |
| | DoLa | **99.5** | 5.7 | **6.7** | 53.8 |
| | ID | 98.3 | 5.0 | 5.7 | 53.5 |
| | ChatProtect | 97.3 | 5.0 | 6.5 | 56.7 |
| | Self-Endorse | 96.7 | 4.1 | 6.3 | 60.6 |
| | Ours-Resp | **99.5** | 3.2 | 5.7 | 64.1 |
| | Ours-Info | 88.5 | 1.9 | 5.9 | 75.6 |
| | Ours-Prec | 78.6 | **1.4** | 5.0 | **78.4** |
| Qwen3-32B | Greedy | **99.5** | 8.8 | 9.7 | 52.4 |
| | DoLa | 95.6 | 9.3 | 8.2 | 53.1 |
| | ChatProtect | 98.9 | 8.1 | 6.8 | 54.4 |
| | Self-Endorse | 91.8 | 4.9 | 8.4 | 63.2 |
| | Ours-Resp | 97.8 | 5.7 | 11 | 65.7 |
| | Ours-Info | 92.9 | 4.2 | **11.7** | 73.5 |
| | Ours-Prec | 82.4 | **3.1** | 11.1 | **78.4** |

for computing semantic entropy, ensuring that existing hallucinations do not affect the identification of hallucinations in newly sampled sentences. The Following strategy is ultimately adopted due to its superior empirical performance, as discussed in Section 5.5.

**Sentence rewriting**. We employ an LLM to rewrite the sentence to remove its hallucinated content while preserving factual information. Specifically, we provide the LLM with a list of factual claims identified by HalluSE and prompt it to generate a sentence comprising those claims, rather than supplying the original sentence along with hallucinated claims and asking it to remove them. Empirically, we find that the former approach performs better with small- to medium-scale models such as Qwen3-32B, Llama3.1-8B, and even Qwen3-4B-Instruct. We hypothesize that this advantage arises because LLMs are more effective when guided by positive examples than by negative ones.

The rewriting LLM can be any model with sufficient instruction-following capability to perform the task. In this work, we use the same model as the main generation model. The rewriting prompts are provided in Appendix A.

## 4.2 ABSTENTION MECHANISM

Our third termination condition leads to a novel dynamic abstention mechanism based on the model's parametric knowledge and internal confidence. Assuming sufficient diversity in sentence sampling, our method abstains after generating $N$ fully hallucinated sentences covering different aspects of the user query in a row. This abstention may occur either at the outset or midway through a generation, with the latter case allowing the model to first produce information it is confident is factual.

Table 2: Performance of the baseline and our methods on the FactScore benchmark with a 200-word response length constraint. Models are prompted to generate approximately 200 words, which exceeds the average length produced without such constraint.

| Model | Method | Response Rate (%) | No. Unsupported | No. Supported | Factual Precision (%) |
|-------|--------|-------------------|-----------------|---------------|----------------------|
| Qwen3-32B | Greedy | **98.9** | 16.2 | 22.4 | 58.0 |
| | DoLa | 97.8 | 16.9 | 22.3 | 56.9 |
| | ChatProtect | 97.8 | 14.7 | 21.3 | 59.2 |
| | Ours-Info | 98.4 | 11.8 | **29.1** | 71.0 |
| | Ours-Prec | 84.6 | **6.7** | 23.6 | **77.9** |

## 5 RESULTS

### 5.1 REDUCED HALLUCINATIONS AND RAISED SUPPORTED FACTS

**Experiment setup.** We mainly evaluate our method on the FactScore benchmarks using Qwen3-32B (Yang et al., 2025a) and Llama3.1-8B-Instruct (Grattafiori et al., 2024). Qwen3-32B follows officially recommended decoding settings with temperature 0.7, top-$k$ 20, and top-$p$ 0.8, while Llama3.1-8B-Instruct uses temperature 0.7, top-$k$ 50, and top-$p$ 0.9. For baselines, we use Greedy decoding, DoLa (Chuang et al., 2024), ID (Cheng et al., 2025b), ChatProtect (Mündler et al., 2024), and Self-Endorse (Wang et al., 2024). The full experiment setup and the configuration of our methods are given in Appendix B.1.

Factual precision is defined as the proportion of supported claims ("No. Supported") relative to the total number of claims ("No. Supported" + "No. Unsupported"). Response rate denotes the proportion of queries answered without refusal. Factual precision and the number of fact claims are computed with generations that answer the prompt queries without refusal.

For each model, results of our method are reported under three hyperparameter settings: Ours-Resp maximizing the response rate, Ours-Info maximizing the number of supported claims ("No. Supported"), and Ours-Prec maximizing factual precision.

**Reduced hallucination rate**. As shown in Tabs. 1 and 2, our method substantially reduces hallucination rates across different models and generation lengths. It consistently improves factual precision over the Greedy baseline by approximately 20–26% and significantly decreases the number of unsupported fact claims that are hallucinated by the model.

**Increased factual information**. In addition, Tabs. 1 and 2 show that our method increases the number of supported fact claims across all setups with Qwen3-32B, indicating that the generated responses contain more factual information and are thus more helpful. For Llama3.1-8B-Instruct, our method slightly reduces supported fact claims, but this is minor compared to the substantial reduction in hallucinated claims.

**Abstention**. We observe that our method achieves the highest factual precision and the largest number of supported facts, albeit with a lower response rate. This indicates that the method effectively identifies user queries for which the underlying model has limited knowledge and abstains from answering. To further validate its effectiveness in mitigating hallucinations independent of additional abstention, we report results under a matched response rate with the baseline, denoted Ours-Resp in Tab. 1 and Ours-Info in Tab. 2. Even under this setting, our method substantially improves factual precision compared to the baseline.

### 5.2 COMPLEMENTARY TO TRAINING-TIME METHODS

The results in Table 3 show that our method provides strong complementary benefits to the training-time hallucination mitigation method FactAlign (Huang & Chen, 2024). Under both unconstrained and length-constrained settings, adding Ours-Resp consistently reduces unsupported claims and improves factual precision, while Ours-Prec achieves the largest gains—boosting precision from 53.1% to 80.6% without length constraints and from 55.4% to 79.1% with a 200-word limit. These re-

Table 3: Performance of combining FactAlign with our method on the FactScore benchmark for the Llama3-8B-Instruct model.

| Gen Length Constraint | Method | Response Rate (%) | No. Unsupported | No. Supported | Factual Precision (%) |
|---|---|---|---|---|---|
| No | FactAlign | **100.0** | 4.2 | **4.7** | 53.1 |
| | + Ours-Resp | 98.4 | 1.7 | 3.7 | 69.1 |
| | + Ours-Prec | 73.6 | **0.9** | 3.6 | **80.6** |
| 200 words | FactAlign | **100.0** | 13.6 | **16.9** | 55.4 |
| | + Ours-Resp | **100.0** | 8.6 | 14.6 | 62.9 |
| | + Ours-Prec | 78.6 | **4.0** | 15.3 | **79.1** |

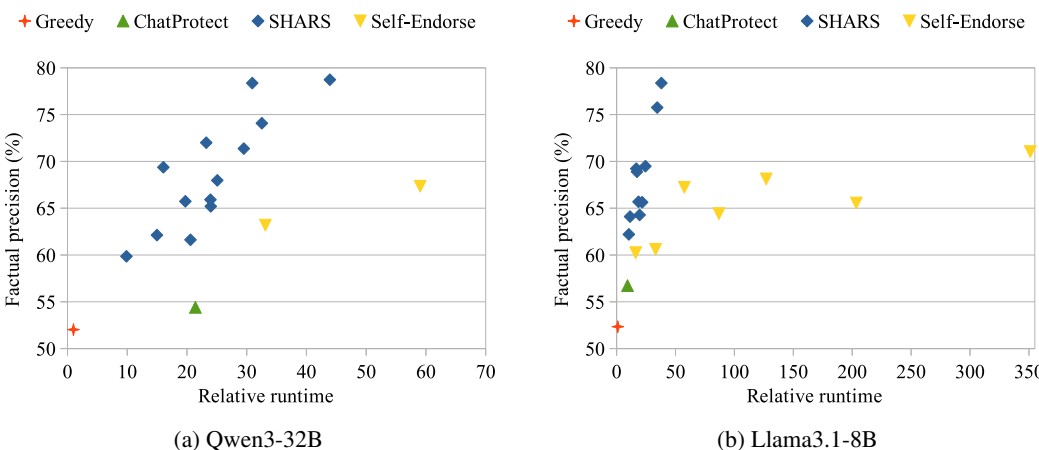

(a) Qwen3-32B  (b) Llama3.1-8B

Figure 3: The efficiency of the baselines and our method on the FactScore benchmark.

sults indicate that combining FactAlign with our method produces a more effective and reliable hallucination-mitigation strategy than FactAlign alone.

## 5.3 EFFICIENT SCALING OF FACTUAL PRECISION

Across both figures in Fig. 3, SHARS consistently demonstrates a far better accuracy–efficiency trade-off than the baselines. In the Qwen3-32B plot, SHARS points cluster in the region of relatively low runtime (roughly 10–40×) while achieving higher factual precision (around 60–78%). In contrast, Self-Endorse requires much higher runtime (about 35–60x) to reach similar or even lower precision, and ChatProtect incurs additional computational cost with only limited improvement over the Greedy baseline. The trend is even more pronounced for Llama3.1-8B: SHARS maintains strong precision (around 68–78%) at runtimes below 50x, whereas Self-Endorse pushes beyond 350x runtime to achieve comparable accuracy of 71%. Taken together, the results show that SHARS delivers higher precision at substantially lower computational cost, making it significantly more efficient than existing methods.

## 5.4 LONG-FORM HALLUCINATION DETECTION

In this section, we compare our method against long-form semantic entropy and other closely related hallucination detection baselines which requires no training and external gold knowledge base.

**Experiment setup.** We evaluate our hallucination detection method without SHARS applied for Qwen3-32B Yang et al. (2025a) on the FactualBio dataset introduced by Farquhar et al. (2024). FactualBio contains paragraph-length biographies of 21 individuals sampled from the WikiBio dataset Lebret et al. (2016). Each paragraph-length biography in the FactualBio dataset is broken down into individual sentences, which are labeled True, Incorrect-Minor, or Incorrect-Major,

Table 4: Performance of the baseline and our methods on the FactualBio benchmark with Qwen3-32B when detecting both Major and Minor hallucinations.

| Model | Method | AUROC | AURAC | Accuracy @ 0.8 | Accuracy @ 0.9 |
|---|---|---|---|---|---|
| Qwen3-32B | Self-Check | 57.6 | 69.3 | 73.5 | **73.5** |
| | P(True) | 69.8 | 73.3 | 70.0 | 70.0 |
| | Naive Long-Form SE | 66.2 | 73.1 | 70.5 | 70.5 |
| | Ours | **72.9** | **77.3** | **75.4** | 72.8 |

Table 5: Performance of the baseline and our methods on the LongFact benchmark under different generation length constraints.

| Model | Gen Length Constraint | Method | Response Rate (%) | No. Unsupported | No. Supported | Factual Precision (%) |
|---|---|---|---|---|---|---|
| Qwen3-32B | No | Baseline | 100.0 | 1.7 | **23.1** | 93.0 |
| | | Ours | 100.0 | **1.1** | 21.2 | **94.6** |
| | 200 words | Baseline | 100.0 | 3.2 | **43.4** | 93.0 |
| | | Ours | 100.0 | **2.5** | 41.8 | **94.4** |

depending on the severity of the false claim. For example, the claim that an individual was knighted, though they were not, is considered Incorrect-Major, while a reported birthdate in the wrong month is considered Incorrect-Minor. The same answers generated by GPT-4 were evaluated for different detection methods.

To benchmark our hallucination detection method, we extended the FactualBio dataset to include "entities", with respect to which our method evaluates the semantic uncertainty of each claim. The Self-Check baseline, rather than evaluating semantic uncertainty, simply asks the LLM whether the factoid is likely to be true. The P(True) baseline considers the probability that the LLM predicts that the next token is "True" when few-shot prompted to compare the original answer with plausible alternatives.

**Improved detection accuracy.** As shown in Table 4, we observe that our method improves hallucination detection AUROC significantly. AUROC measures how well the uncertainty score distinguishes correct from incorrect answers across all thresholds. AURAC, or the area under the 'rejection accuracy' curve, summarizes how much accuracy improves when discarding the most uncertain answers. Accuracy@0.8 and Accuracy@0.9 report the model's accuracy after discarding the top 20% and top 10% most uncertain responses, respectively.

## 5.5 ABLATION STUDY

This section presents an ablation study on two components of our method: sentence sampling and rewriting. As shown in Tab. 6 in Appendix B.3, both rewriting and the Following sampling strategy are critical for achieving strong performance. Enabling rewriting substantially boosts the response rate, while the Following strategy increases the number of supported fact claims and, when combined with rewriting, further improves factual precision.

## 5.6 ADDITIONAL RESULTS ON LONGFACT BENCHMARK

In addition to FactScore, we evaluate our method on an alternative long-form factuality benchmark, LongFact (Wei et al., 2024). As shown in Tab. 6, our method consistently mitigates hallucinations on LongFact, improving factual precision and reducing unsupported fact claims compared to the baseline. Although the improvement margin is smaller than in the FactScore experiments, this is expected since the baseline already achieves very high factual precision. Notably, the 1.4% precision gain from our method is comparable to the 0.9% improvement observed when moving from GPT-3.5-Turbo to GPT-4-Turbo, as reported in Wei et al. (2024). We emphasize that the reported results

are based on a single run without hyperparameter tuning due to the high API cost of evaluation, suggesting that further performance gains are likely achievable with hyperparameter optimization.

# 6 LIMITATIONS

Our approach requires substantial inference-time compute, which increases cost and limits its practicality in resource-constrained settings. Furthermore, reliance on instruction-following means that it cannot always be effectively applied to small-scale models that do not have sufficient instruction-following capabilities. Future work will explore integrating lightweight semantic probes or developing purpose-built smaller models fine-tuned for fact decomposition, rewriting, and question generation, which could broaden the applicability and reduce computational demands.

# 7 CONCLUSION

In conclusion, this work addresses the critical challenge of hallucinations in open-ended generation by introducing SHARS, a general inference-time compute framework that incrementally rejects hallucinated content and builds subsequent outputs upon verified information. Together with HalluSE, our improved uncertainty-based detection method, SHARS provides an effective and flexible approach to mitigating hallucinations while maintaining or enhancing informativeness. Extensive evaluations across multiple long-form factuality benchmarks demonstrate that our methods significantly advance the state of hallucination detection and mitigation, and importantly, reveal a promising inference-time scaling property of factuality. These findings highlight the potential of inference-time compute as a powerful and practical paradigm for improving the reliability of large language models, especially in high-stakes domains where accuracy and trustworthiness are paramount.

## REPRODUCIBILITY STATEMENT

Upon acceptance, we will release our code, appropriate environment builders, and all configurations to facilitate reproduction of our results. We will also provide dataloaders for all datasets used in this work. Because generation involves non-deterministic sampling from LLMs, we cannot guarantee identical outputs across runs; however, we will ensure that all experimental protocols and hyperparameters are well-documented so that results can be faithfully approximated.

## ETHICS STATEMENT

In conducting this research, we commit to the guiding principles outlined in the ICLR Code of Ethics. Our goal is to contribute positively to society and the field of language model reasoning, with a particular focus on mitigating the harms posed by LLM hallucinations. We are conscious that our method relies on more intensive compute demands, which carry environmental costs that may disproportionately affect climate-insecure communities. We hope to minimize this in future work by developing smaller models that are able to perform the same tasks. We also acknowledge that our experiments rely on biographical information from open-source datasets. While such data may raise privacy considerations, we neither collect nor use private or personal data without consent, and all evaluations are based on publicly available resources and model generations. We assess that this use does not compromise individual privacy. Finally, we refrain from overstating claims or hiding negative results. We encourage users of our method to perform risk assessments before deployment in sensitive domains, such as medicine.

## LLM STATEMENT

Beyond being the subject of our experiments, LLMs were used in this work to write figure-generating code, format LaTeX tables, perform sentence-level clarity edits on some sections of text, and write some utility functions in our codebase (e.g., "make me a function that converts 'three' to 3.").

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

# A    PROMPTS

Prompts are given in Fig. 4 and Fig. 5.

# B    EXPERIMENTS

## B.1    CONFIGURATION

FactScore (Min et al., 2023) evaluates LLMs by generating biographies for 182 individuals [1] (the labeled split), spanning diverse demographics and varying levels of rarity. Each generation is decomposed into atomic facts, which are verified against a reliable knowledge source, in this case a pre-saved Wikipedia. GPT-5 was used as the backend LLM for the benchmark. The prompts for query are given in Appendix A. The results are reported from a single run due to the high API cost of benchmark evaluation.

LongFact (Wei et al., 2024) is a benchmark for long-form factuality with two key differences from FactScore. First, it includes thousands of questions across 38 topics, though we used only a subset of 140 prompts following Zhao et al. (2025) due to resource constraints (evaluating a single generation costs at least $0.19 (Wei et al., 2024)). Second, it relies on results from online Google Search rather than a pre-saved Wikipedia as the knowledge source. GPT-3.5-turbo-0125 is used as the backend LLM. The experimental setup follows that of FactScore in Section 5.1.

We describe here the hyperparameters of our method. The maximum number of tolerated consecutive hallucinated sentences sampling, $N$, is 10 across all setups. In Tab. 1, the number of probe questions, $Q$, the number of answers, $A$, and the semantic entropy threshold, $\theta$, are 1, 3, 0.7 for Ours-Resp with Llama3.1-8B-Instruct; 3, 3, 0.3 for Ours-Info with Llama3.1-8B-Instruct; 2, 3, 0.3 for Ours-Prec with Llama3.1-8B-Instruct; 1, 5, 0.7 for Ours-Resp with Qwen3-32B; 2, 3, 0.2 for Ours-Info with Qwen3-32B; 2, 7, 0.6 for Ours-Prec with Qwen3-32B. In Tab. 2, the number of probe questions, $Q$, the number of answers, $A$, and the semantic entropy threshold, $\theta$, are 1, 3, 0.5 for Ours-Info; 2, 3, 0.3 for Ours-Prec. In Tab. 5, the number of probe questions, $Q$, the number of answers, $A$, and the semantic entropy threshold, $\theta$, are 3, 3, 0.3 for without length constraint; 2, 3, 0.5 for 200-words constraint. In Tab. 6, the number of probe questions, $Q$, the number of answers, $A$, and the semantic entropy threshold, $\theta$, are 1, 3, 0.5 for all.

All above hyperparameters, except for the ones for LongFact, are found through a coarse grid search.

## B.2    NLI MODEL IN SEMANTIC ENTROPY

The NLI model we used is DeBERTa V2 (He et al., 2021) with 900M parameters, identical to the one used in Semantic Entropy. Specifically, we used the deberta-v2-xlarge-mnli checkpoint from HuggingFace. It was trained on the following datasets: Wikipedia (English dump; 12GB), BookCorpus (Zhu et al., 2015; 6GB), OPENWEBTEXT (public Reddit content, Gokaslan & Cohen, 2019; 38GB), and STORIES (a CommonCrawl subset, Trinh & Le, 2018; 31GB). These sources span diverse domains, making the model effectively domain-agnostic.

## B.3    RESULTS OF ABLATION STUDY

The results of our ablation study are given in Tab. 6.

---

[1]We exclude one individual named Focus... from the original dataset due to confusion with the band of the same name Focus and complications caused by special punctuation.

Table 6: Performance of various variants of our method on the FactScore benchmark for Qwen3-32B model. No generation length constraint was applied. All variants were evaluated with the same hyperparameter settings described in Appendix B.1. Relative runtime is reported as a factor with respect to the runtime of the Following-Rewrite variant.

| Sentence Sampling | Rewrite | Response Rate (%) | No. Unsupported | No. Supported | Factual Precision (%) | Relative Runtime |
|---|---|---|---|---|---|---|
| Following | Yes | 91.8 | 4.8 | 10.7 | 69.4 | **1.00** |
| Temperature | Yes | **95.6** | 4.9 | 9.0 | 64.8 | 1.01 |
| Following | No | 54.4 | 4.3 | **12.0** | 73.5 | 1.60 |
| Temperature | No | 40.1 | **2.3** | 7.4 | **76.2** | 1.55 |

**Legend**  **System Prompt**  **User Prompt**

## Main Prompt

You are given a query from the user asking for information. Write the answer in English characters. Output plain text only. Do not use formatting styles, symbols, or headings. Each sentence should provide different information and must not repeat the same content. Output only the final answer to the query. Do not generate explanations, reasoning, or commentary for the answer. Do not ask follow-up questions. The response must be around 200 words in total.

Tell me about {entity}.

## Question Generation Prompt

You are given a query, a sentence and an entity from that sentence. The sentence is the part of a generated response to the given query. Your task is to generate 1 distinct, natural questions such that:
1. The given sentence serves as a full long-form answer.
2. The given entity serves as a correct short-form answer.

### Instructions:

* Each question must be open-ended (cannot be answered by yes/no).
* Do not mention or hint the answer in the question.
* Each question must be phrased differently (no redundancy, varied structures).

### Output Format:

* output the questions as a list
* each line must follow this format: `- question`

----

### Example 1

#### Input
Query: Tell me about the Eiffel Tower.
Sentence: The Eiffel Tower in Paris was completed in 1889.
Entity: 1889

#### Output
- In which year was the Eiffel Tower in Paris completed?

### Example 2

#### Input
Query: Tell me about Davy Crockett.
Sentence: Davy Crockett was a frontiersman.
Entity: frontiersman

#### Output
- What was Davy Crockett's profession?

Query: {query}.
Sentence: {claim}
Entity: {entity}

## Answer Generation Prompt

You are given from the user a question, along with a query and verified information as context. Answer the question only in plain text. Do not use full sentences. Respond with the fewest words possible, such as a name, place, or thing. If multiple valid answers exist, output up to 5 answers, separated by `;`.

Query: {query}
Verified information: {verified_text}
Question: {question}

## Fact Decomposition Prompt

You are an information extraction assistant. Your task is to analyze a sentence and output a list of entities and their corresponding fact claims.

---

### Task Instructions

1. Identify all entities in the sentence, excluding the subject.
2. For each entity, generate one fact claim about it based only on the sentence.

---

### Rules

#### Entity Rules:

* Entities include named persons, occupations, organizations, objects, concepts, events, locations, roles, fields, education, times, and numbers.
* Must not include the sentence subject and pronouns like He/She/I/It/This/That/They.
* Copy entities exactly as written in the sentence (no rephrasing, no normalization).

#### Fact Claim Rules:

* Each entity must map to exactly one fact claim.
* A fact claim must:
  - Be a single, complete, grammatically correct sentence.
  - Be fully supported by the input sentence (no added or inferred information).
  - Contain the entity itself explicitly.
* Each fact claim should include only the target entity, unless other entities are strictly necessary to preserve the meaning.

#### Output Format:

* If no entities remain after exclusion, return an empty list.
* Otherwise, return as a list.
* Each line must follow the format: `- entity: fact claim`

---

### Example 1

Input: Isaac Newton was an English mathematician and physicist.

Output:
- Chinese: Isaac Newton was English.
- researcher: Isaac Newton was a mathematician.
- writer: Isaac Newton was a physicist.

---

### Example 2

Input: Marie Curie won Nobel Prizes in both Physics and Chemistry.

Output:
- Nobel Prizes: Marie Curie won Nobel Prizes.
- Physics: Marie Curie won Nobel Prizes in Physics.
- Chemistry: Marie Curie won Nobel Prizes in Chemistry.

---

### Example 3

Input: He has won several/various/many awards.

Output:
- awards: He has won awards.

{sentence}

Figure 4: Set of prompts for various parts of the pipeline.

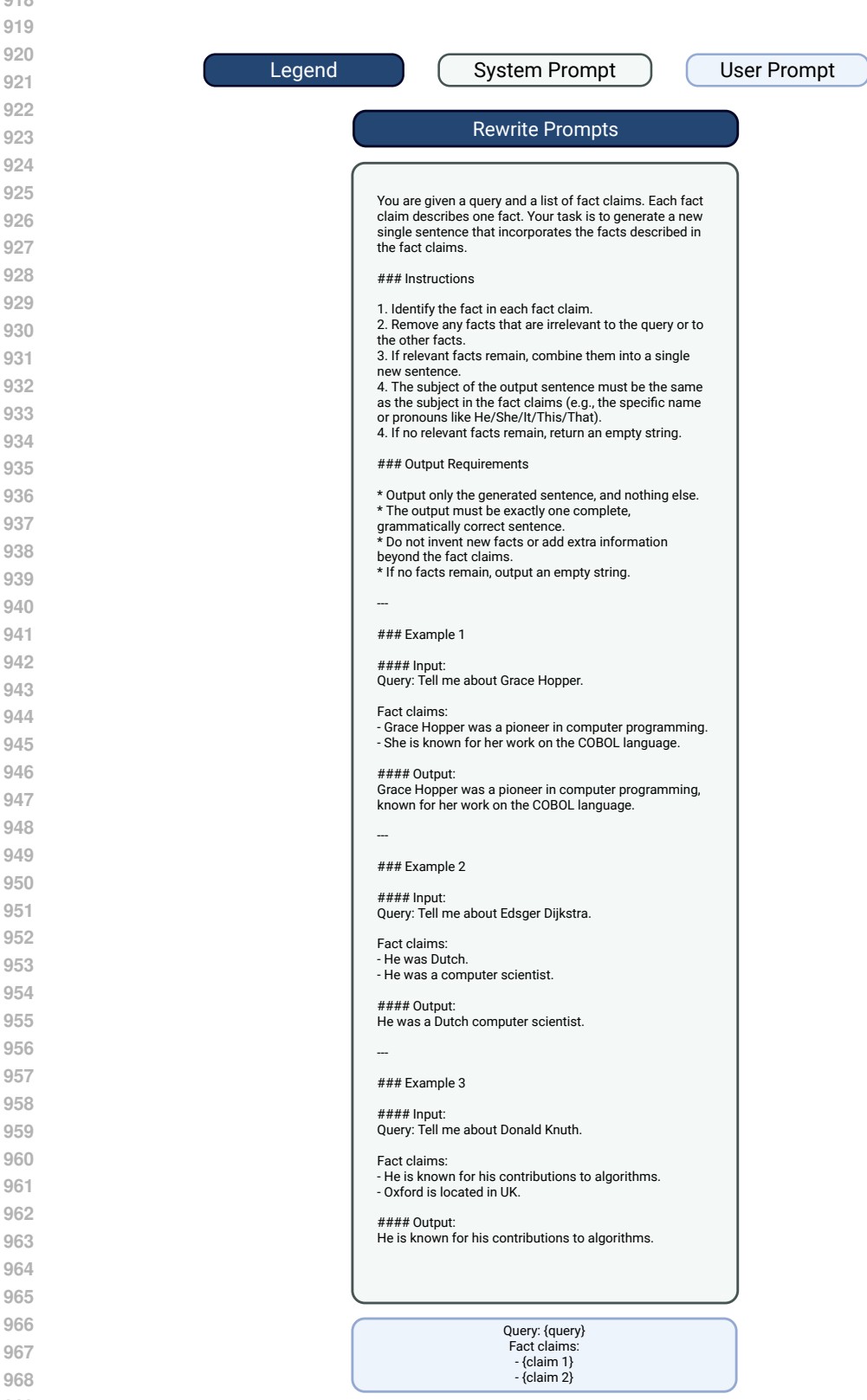

Figure 5: Additional prompt if rewrite is enabled.

