# OpenReview forum: "Building Reliable Long-Form Generation via Step-Wise Hallucination Rejection Sampling"
_ICLR.cc/2026/Conference — Submitted to ICLR 2026_

### Official Review · Reviewer_y9bF · 2025-10-16

**Soundness:** 2
**Presentation:** 2
**Contribution:** 2
**Rating:** 2
**Confidence:** 5

**Summary:**

The paper introduces an inference-time scaling framework (SHARS), aiming to allocate additional computational resources to detect and mitigate hallucinations during decoding.
As a main component of the framework, the uncertainty-based hallucination detection method HalluSE which aims to improve semantic entropy is introduced.
HalluSE is evaluated on long-form hallucination benchmarks, showing improved performance over some baseline methods.
The overall SHARS framework is evaluated by the FactScore benchmark, indicating less unsupported answers and high factual precision, at the cost of decreased response rate, showcasing scaling behavior with increased compute.

**Strengths:**

- The paper tackles an important problem and shows interesting scaling behavior, where more compute leads to higher factual correctness.
- The experiments use recent models (Llama 3.1 and QWEN 3) with both 8B and 32B parameters.
- Experiments tackle hallucinations in long-form generation, which is often omitted in prior work.
- HalluSE aims to improve the performance of semantic entropy in such long-form answer settings

**Weaknesses:**

- It would help to more clearly state that the overall framework depends on an auxiliary NLI model. Line 164 notes that Semantic Entropy (which HalluSE seeks to improve) requires mapping generated sequences to semantic clusters and refers to Farquhar et al. for details; in those works (and in the original work on semantic entropy - Kuhn et al., 2023), this mapping is performed with a BERT-style NLI model. However, the present paper does not discuss the specific NLI model used. This omission is important in light of the claims in lines 293ff that HalluSE "(1) is training-free and domain-agnostic, and (2) does not rely on external models, tools, or reference knowledge sources." Given the dependence on an NLI model, the "training-free", "domain-agnostic" and "no external models" characterizations are not justified if the NLI model's training data is domain-specific. Even when utilizing some a priori trained model as in Farquhar and Kuhn et al., I recommend explicitly describing the NLI component (architecture, checkpoint, and training data) and revising the scope of these claims. This would strengthen the paper's transparency and help readers assess generality. In the current form, I find the presentation not acceptable and misleading.
- The changes to the original semantic entropy algorithm appears minor, in essence it is about different prompting strategies to generate answers rather than any methodological advancement in how to calculate the uncertainty. However, while pretty much ad-hoc without any theoretical grounding, calculating long-form semantic entropy with HalluSE might be more useful than the originally proposed approach to scale Semantic Entropy to long form generation which was arguably suboptimal.
- The text in 5.1 does not clearly mention if only HalluSE is evaluated in this experiments or if the full SHARS framework is used as well to generate the answers.
- Baselines for the experiments reported in Table 1 are very limited, the literature on uncertainty estimation in LLMs is vast and there are many methods such as SAR (Duan at al.) or INSIDE (Chen et al.) that showed very strong results recently and could easily be incorporated.
- Overall, the proposed framework explains a high-level workflow that orchestrates multiple calls to LLMs. While it may be useful for particular practical settings, it relies heavily on the correct working of subcalls for fact decomposition, question generation and answer sampling. For me, the proposed SHARS framework does not pass the bar on profound methodological advancement I would expect from an ICLR paper. It might be better positioned for a venue focused more on applied or empirical studies, such as EMNLP, where the practical insights and orchestration design could be particularly appreciated.

---

Lorenz Kuhn, Yarin Gal, Sebastian Farquhar (2023) Semantic Uncertainty: Linguistic Invariances for Uncertainty Estimation in Natural Language Generation, ICLR

Jinhao Duan, Hao Cheng, Shiqi Wang, Alex Zavalny, Chenan Wang, Renjing Xu, Bhavya Kailkhura, Kaidi Xu (2024) Shifting Attention to Relevance: Towards the Predictive Uncertainty Quantification of Free-Form Large Language Models, ACL

Chao Chen, Kai Liu, Ze Chen, Yi Gu, Yue Wu, Mingyuan Tao, Zhihang Fu, Jieping Ye (2024) INSIDE: LLMs' Internal States Retain the Power of Hallucination Detection, ICLR

**Questions:**

- What is the accuracy @ 1.0 for Table 1? Are the same answers by the QWEN model evaluated for different methods?
- The results in 5.2 are lacking a comparison for computational costs, e.g. simply by avg. walltime for answer generation or by token counts. The authors are upfront that their method incurs additional computational costs, but how much exactly? Similarly, it would be interesting to see such a comparison for the two considered sampling strategies (rewriting and Following). The headline results in Figure 1b suggests they are extreme, like up to 50 times compared to the standard strategy. It would be interesting to have more insight into the distribution of runtimes over a given dataset.

---

> ### Author Response · Authors · 2025-12-03
>
> > It would help to more clearly state that the overall framework depends on an auxiliary NLI model.
>
> Thank you for this valuable suggestion. The NLI model we used is DeBERTa V2 \[1\] with 900M parameters, identical to the one used in Semantic Entropy. Specifically, we used the deberta-v2-xlarge-mnli checkpoint from HuggingFace. It was trained on the following datasets: Wikipedia (English dump; 12GB), BookCorpus (Zhu et al., 2015; 6GB), OPENWEBTEXT (public Reddit content, Gokaslan & Cohen, 2019; 38GB), and STORIES (a CommonCrawl subset, Trinh & Le, 2018; 31GB). These sources span diverse domains, making the model effectively domain-agnostic. We have added this description to Appendix B.2 in the revised manuscript.
>
> Regarding characterizations, our intent was to highlight that, unlike some hallucination detectors like Linear Probe, the adopted detector does not require training a new model. We have revised the relevant description in the manuscript to improve clarity.
>
> > The changes to the original semantic entropy algorithm appears minor, in essence it is about different prompting strategies to generate answers rather than any methodological advancement in how to calculate the uncertainty. However, while pretty much ad-hoc without any theoretical grounding, calculating long-form semantic entropy with HalluSE might be more useful than the originally proposed approach to scale Semantic Entropy to long form generation which was arguably suboptimal.
>
> We appreciate the reviewer’s recognition of HalluSE’s utility in long-form hallucination detection, and acknowledge the reviewer's comments regarding its limitations. Nevertheless, we wish to emphasize that our primary methodological contribution resides in the inference-time scaling method, SHARS, rather than hallucination detection or uncertainty quantification.
>
> > The text in 5.1 does not clearly mention if only HalluSE is evaluated in this experiments or if the full SHARS framework is used as well to generate the answers.
>
> Only HalluSE is evaluated in Section 5.1 in the initial manuscript (Section 5.4 in the revised manuscript). As stated in the experimental setup, this section focuses specifically on “evaluate our hallucination detection method.” To improve clarity, we have revised the manuscript to explicitly note that SHARS is not used in this evaluation.
>
> > Baselines for the experiments reported in Table 1 are very limited, the literature on uncertainty estimation in LLMs is vast and there are many methods such as SAR (Duan at al.) or INSIDE (Chen et al.) that showed very strong results recently and could easily be incorporated.
>
> We appreciate the reviewer’s pointers to relevant literature and have added a dedicated description in Section 5.1 to address these references. However, we emphasize that **uncertainty estimation, while valuable, is not the focus of this work, nor do we claim state-of-the-art performance in that area**.
>
> Our primary contribution is an inference-time scaling framework for mitigating hallucinations in long-form text generation. The framework relies on an arbitrary hallucination detector to identify hallucinations during decoding. In this paper, **we adopt an improved semantic entropy pipeline as the specific detector, but as noted, it can be replaced with any more advanced alternative.** We leave this exploration for future work, as the current implementation already achieves substantial reductions in hallucinations and significant improvements in factual precision compared with existing state-of-the-art hallucination mitigation methods (see the *Shared Response* at the top of this page).

---

> ### Author Response · Authors · 2025-12-03
>
> > Overall, the proposed framework explains a high-level workflow that orchestrates multiple calls to LLMs. While it may be useful for particular practical settings, it relies heavily on the correct working of subcalls for fact decomposition, question generation and answer sampling. For me, the proposed SHARS framework does not pass the bar on profound methodological advancement I would expect from an ICLR paper. It might be better positioned for a venue focused more on applied or empirical studies, such as EMNLP, where the practical insights and orchestration design could be particularly appreciated.
>
> We thank the reviewer for acknowledging the utility of our method and the practical insights it provides. **Regarding the concern about reliance on successful subcalls, we have shown that they remain reliable across diverse model scales**, as demonstrated by our method’s strong performance even with compact models such as Qwen3-4B as shown in Table 1 in the revised manuscript.
>
> In response to the concern that our practical insights and orchestration design may not meet the bar for substantial methodological advancement, we emphasize that **the core of our contribution is a novel decoding strategy rather than an orchestration mechanism**, as illustrated in Algorithms 2 and 3. SHARS integrates two principal components: real-time hallucination detection and filtering, and a semantic diversity–enhanced sampling strategy.
>
> We acknowledge that expectations for ICLR publications may vary across reviewers. We would like to note, however, that **many works with a similar methodological design philosophy have been published at ICLR and other top ML venues, such as Self-Consistency [2], ChatProtect [3], and Self-Refine [4]**.
>
> > What is the accuracy @ 1.0 for Table 1? Are the same answers by the QWEN model evaluated for different methods?
>
> We did not use a metric named accuracy@1.0 in Table 1 or elsewhere in the initial manuscript. We assume the reviewer is referring to accuracy@0.8/0.9 in Table 1, which is explained at Line 377 on page 7 of the initial manuscript.
>
> Yes, the same answers were evaluated for different detection methods, but the answers were generated by GPT-4 rather than Qwen, as provided in the adopted dataset. We have revised the manuscript to explain this.
>
> > The results in 5.2 are lacking a comparison for computational costs, e.g. simply by avg. walltime for answer generation or by token counts. The authors are upfront that their method incurs additional computational costs, but how much exactly? Similarly, it would be interesting to see such a comparison for the two considered sampling strategies (rewriting and Following). The headline results in Figure 1b suggests they are extreme, like up to 50 times compared to the standard strategy. It would be interesting to have more insight into the distribution of runtimes over a given dataset.
>
> The suggested comparison of methods in Section 5.2 is provided in Figure 1b of the initial manuscript. The comparison of the two sampling strategies appears in Table 5 of the initial manuscript (Table 6 in the revised version). We also refer the reviewer to the Shared Response for a detailed efficiency analysis.
>
> *References*:
> 1. He, Pengcheng, et al. "Deberta: Decoding-enhanced bert with disentangled attention." arXiv preprint arXiv:2006.03654 (2020).
> 2. Wang, Xuezhi, et al. "Self-Consistency Improves Chain of Thought Reasoning in Language Models." The Eleventh International Conference on Learning Representations. 2023.
> 3. Mündler, Niels, et al. "Self-contradictory Hallucinations of Large Language Models: Evaluation, Detection and Mitigation." The Twelfth International Conference on Learning Representations. 2024
> 4. Madaan, Aman, et al. "Self-refine: Iterative refinement with self-feedback." Advances in Neural Information Processing Systems 36 (2023): 46534-46594.

---

### Official Review · Reviewer_daAC · 2025-10-28

**Soundness:** 2
**Presentation:** 2
**Contribution:** 1
**Rating:** 2
**Confidence:** 5

**Summary:**

The submission introduces SHARS/HalluSE - Step-wise HAllucination Rejection Sampling, a method that is intended to address some classes of the hallucinations by resampling the outputs that have high semantic entropy.
The method uses a multistage pipeline that breaks down the model outputs into atomic facts, formulates them as subquestions, samples and evaluates semantic entropy (SE) on each of them individually.
This improves the hallucination detection on test data compared to regular SE.

**Strengths:**

The authors consider the problem of applying semantic entropy to longer generations.

**Weaknesses:**

1. A large volume of state of the art uncertainty estimation literature is ignored, both in writing and in evaluation ( e.g. [1,2,3]).
2. The method consists mostly of prompt engineering.
3. Limited evaluation, the results are presented for two models and two datasets, which is way below the standard of the field even if slack is given for compute availability ([4]).
4. Theoretical justification is absent. The authors spam algorithms, yet fail to derive any identities for what they are doing (even though they are mentioning Rejection Sampling, which could be quite interesting, even the separation of the output into facts could be treated in an interesting fashion).

### References
1. Aichberger, L., Schweighofer, K. & Hochreiter, S. Rethinking Uncertainty Estimation in Natural Language Generation. Preprint at https://doi.org/10.48550/arXiv.2412.15176 (2024).
2. Duan, J. et al. Shifting Attention to Relevance: Towards the Uncertainty Estimation of Large Language Models. Preprint at https://doi.org/10.48550/arXiv.2307.01379 (2023).
3. Manakul, P., Liusie, A. & Gales, M. J. F. SelfCheckGPT: Zero-Resource Black-Box Hallucination Detection for Generative Large Language Models. Preprint at https://doi.org/10.48550/arXiv.2303.08896 (2023).
4. Ielanskyi, M., Schweighofer, K., Aichberger, L. & Hochreiter, S. Addressing Pitfalls in the Evaluation of Uncertainty Estimation Methods for Natural Language Generation. Preprint at https://doi.org/10.48550/arXiv.2510.02279 (2025).

**Questions:**

1. Could this method generalize to problems that do not have a very specific 'fact' structure? For example generation of code or tool calls?
2. What happens if the model is still confident about hallucinated fact (i.e. low semantic entropy) after the "Hallucination Identification" step.

---

> ### Author Response · Authors · 2025-12-03
>
> > A large volume of state of the art uncertainty estimation literature is ignored, both in writing and in evaluation ( e.g. \[1,2,3\]).
>
> We appreciate the reviewer’s pointers to relevant literature and have added a dedicated description in Section 5.1 to address these references. However, we emphasize that **uncertainty estimation, while valuable, is not the focus of this work, nor do we claim state-of-the-art performance in that area**.
>
> Our primary contribution is an inference-time scaling framework for mitigating hallucinations in long-form text generation. The framework relies on an arbitrary hallucination detector to identify hallucinations during decoding. In this paper, **we adopt an improved semantic entropy pipeline as the specific detector, but as noted, it can be replaced with any more advanced alternative.** We leave this exploration for future work, as the current implementation already achieves substantial reductions in hallucinations and significant improvements in factual precision compared with existing state-of-the-art hallucination mitigation methods (see the *Shared Response* at the top of this page).
>
> > The method consists mostly of prompt engineering.
>
> **The core of our method is a novel decoding strategy**, not prompt engineering, as illustrated in Algorithms 2 and 3\. The proposed SHARS approach primarily integrates two core mechanisms: real-time hallucination detection and filtering, and a semantic diversity-enhanced sampling strategy.
>
> **We contest the assertion that "consisting mostly of prompt engineering" constitutes a valid critique of a method**. Numerous influential works, such as chain-of-thoughts and in-context learning for general capability, and Self-Endorse and ChatProtect for hallucination mitigation, heavily or even solely rely on prompting. Nevertheless, this does not diminish their profound contribution to their respective domains.
>
>  > Limited evaluation, the results are presented for two models and two datasets, which is way below the standard of the field even if slack is given for compute availability (\[4\]).
>
> We note that \[4\] (referred by the reviewer) primarily reviews uncertainty estimation evaluation, which is outside the scope of the problem we address, namely hallucination in long-form text generation. **For hallucination evaluation, our experimental setup is consistent with those used in established, published work.** For instance, FactTune \[1\] employed two datasets and two models (LLaMA 1 and 2), while ChatProtect used one dataset and four models.
>
> To further strengthen our evaluation, we have added results for an additional model, Qwen3–4B, in Table 1, and included a new setup involving FactAlign—a model trained specifically to improve factuality—in Table 3 of the revised manuscript. Across all settings, our method consistently achieves substantial reductions in unsupported claims and significant gains in factual precision.
>
> > Theoretical justification is absent. The authors spam algorithms, yet fail to derive any identities for what they are doing (even though they are mentioning Rejection Sampling, which could be quite interesting, even the separation of the output into facts could be treated in an interesting fashion).
>
> A theoretical justification is not a prerequisite for all valuable research; indeed, numerous well-established, closely related works in inference-time scaling \[2, 3\] and hallucination mitigation like ChatProtect and Self-Endorse have been published without such a basis. We contend that **our proposed methods, HalluSE and SHARS, are adequately justified by elucidating the mechanisms underlying their efficacy**. HalluSE specifically addresses the limitations inherent in naive long-form semantic entropy, as detailed in Section 3.2 and 3.3. SHARS, conversely, mitigates hallucinations through their detection and subsequent filtering, thereby interrupting the propagation of compounding error throughout the long-form generation process.
>
> > Could this method generalize to problems that do not have a very specific 'fact' structure? For example generation of code or tool calls?
>
> Our method readily generalizes to non-factual hallucination errors, such as the hallucination of a tool or function name. Furthermore, beyond addressing hallucination, our method retains the potential for generalization to non-hallucination errors. To achieve this, one should replace the current instantiation's hallucination detector with a detector tailored to the target error.
>
> > What happens if the model is still confident about hallucinated fact (i.e. low semantic entropy) after the "Hallucination Identification" step.
>
> This information will be retained in the generated output due to a limitation of the adopted hallucination detector. However, the substantial empirical improvement demonstrated by our method suggests that a large proportion of hallucinations do not fall into the category of such "confidently wrong" cases.

---

> > ### Author Response · Authors · 2025-12-03
> >
> > *References*:
> >
> > 1. Tian, Katherine, et al. "Fine-tuning language models for factuality." *The Twelfth International Conference on Learning Representations*. 2024.
> > 2. Wang, Xuezhi, et al. "Self-Consistency Improves Chain of Thought Reasoning in Language Models." *The Eleventh International Conference on Learning Representations*. 2023.
> > 3. Muennighoff, Niklas, et al. "s1: Simple test-time scaling." *Proceedings of the 2025 Conference on Empirical Methods in Natural Language Processing*. 2025.

---

### Official Review · Reviewer_MR2W · 2025-10-30

**Soundness:** 2
**Presentation:** 3
**Contribution:** 2
**Rating:** 4
**Confidence:** 5

**Summary:**

This paper addresses the "hallucination snowballing" effect in long-form generation, where early factual errors propagate and degrade overall reliability. The authors propose a novel inference-time framework, Step-wise HAllucination Rejection Sampling (SHARS), which operates incrementally at the sentence level. Instead of post-hoc verification, SHARS assesses each new sentence for factuality as it is generated. Hallucinated sentences are either discarded or rewritten, ensuring that subsequent generation is conditioned only on verified content. To enable this, the authors introduce HalluSE, an improved uncertainty-based hallucination detector that refines prior semantic entropy methods. A key feature is that the system is self-contained, not requiring external knowledge sources, though it remains compatible with them. Extensive experiments on benchmarks like FactScore and LongFact demonstrate that SHARS significantly reduces hallucinations and improves factual precision, often while increasing the total amount of supported factual information.

**Strengths:**

1. The method is supported by compelling empirical evidence across multiple benchmarks (FactScore, LongFact) and models (Llama3, Qwen3). The results consistently show that SHARS significantly improves factual precision and reduces the number of unsupported claims.
2. The paper is well-written and clearly structured, making the complex methodology easy to follow and understand.

**Weaknesses:**

1. My primary concern is the substantial increase in computational cost and latency at inference time. Each sentence requires a multi-step process of decomposition, question generation, sampling, and potential rewriting, making it significantly slower than naive decoding. This overhead may be prohibitive for real-time or resource-constrained applications.
2. The entire pipeline is highly dependent on the instruction-following capabilities of the base LLM. As the authors note, the framework's effectiveness diminishes with smaller or less capable models that struggle to perform the complex auxiliary tasks required.
3. The framework introduces several new hyperparameters (e.g., number of probe questions Q, sampled answers A, entropy threshold θ) that require careful tuning. The paper shows that optimal settings vary across models and objectives, suggesting that adapting the method to new use cases could be a complex and costly process.
Overall, the proposed method trades significant inference efficiency for improved hallucination mitigation. Given the complexity of the pipeline, I have reservations about its practical feasibility in real-world scenarios.

**Questions:**

Refer to Weaknesses

---

> ### Author Response · Authors · 2025-12-03
>
> > My primary concern is the substantial increase in computational cost and latency at inference time. Each sentence requires a multi-step process of decomposition, question generation, sampling, and potential rewriting, making it significantly slower than naive decoding. This overhead may be prohibitive for real-time or resource-constrained applications.
>
> All inference-time scaling methods introduce some computational overhead. In many real-world settings—such as medical diagnosis or education—users are willing to wait longer for more accurate responses. Thus, the key question is not whether additional computation is introduced, but how much performance gain it yields and how efficiently it compares to alternative methods. **Our approach delivers substantial improvements in factual precision while requiring significantly less computational budget than competitive methods.** We kindly refer the reviewer to the Shared Response for the detailed efficiency analysis.
>
> > The entire pipeline is highly dependent on the instruction-following capabilities of the base LLM. As the authors note, the framework's effectiveness diminishes with smaller or less capable models that struggle to perform the complex auxiliary tasks required.
>
> Our method demonstrates strong performance with small-scale models. In our initial evaluation, we tested with LLaMA 3.1–8B and Qwen3-32B and observed substantial improvements in factual precision over the baselines (+15–26%). To further validate its effectiveness on smaller models, **we additionally evaluated with Qwen3-4B and achieved similar performance gains (+16–24%)**, as shown in Table 1 of the revised manuscript. We believe that models with more than 4B parameters already represent a broad range of architectures used in both academic research and real-world applications.
>
> > The framework introduces several new hyperparameters (e.g., number of probe questions Q, sampled answers A, entropy threshold θ) that require careful tuning. The paper shows that optimal settings vary across models and objectives, suggesting that adapting the method to new use cases could be a complex and costly process. Overall, the proposed method trades significant inference efficiency for improved hallucination mitigation. Given the complexity of the pipeline, I have reservations about its practical feasibility in real-world scenarios.
>
> **Our method can deliver substantial performance improvements across diverse models and datasets using fixed hyperparameter values**, and these gains can be further enhanced by tuning the hyperparameters for specific models or data. For example, we find that Q=3, A=3, and theta=0.3 consistently achieve the highest or near-highest factual accuracy across all settings, while Q=1, A=3, and theta=0.5 provide strong accuracy gains with moderate computational cost. Moreover, we do not believe that having three hyperparameters introduces significant complexity; even the naive decoding strategy also allows tuning multiple hyperparameters, such as temperature, top-k, and top-p.

---

### Official Review · Reviewer_frm6 · 2025-11-01

**Soundness:** 2
**Presentation:** 2
**Contribution:** 2
**Rating:** 4
**Confidence:** 3

**Summary:**

This paper introduces Step-wise Hallucination Rejection Sampling (SHARS), an inference-time framework aimed at improving the factual reliability of large language models (LLMs) in long-form generation. The key idea is to allocate additional compute during decoding by detecting and rejecting hallucinated sentences as they are produced, preventing “hallucination snowballing.”

Empirical results on FactualBio, FactScore, and LongFact benchmarks (using Llama3.1-8B-Instruct and Qwen3-32B) demonstrate that SHARS with HalluSE reduces hallucination rates by 20–26% and improves factual precision, with a consistent positive scaling trend with increased inference-time computation.

**Strengths:**

1. Introduces a new inference-time paradigm—step-wise hallucination rejection—for long-form generation.

2. Well-specified algorithms and transparent design choices.

3.  Figures, tables, and prompts (Appendix A–B) are clear and reproducible.

**Weaknesses:**

1. Only two models (Llama3.1-8B, Qwen3-32B) and few baselines are tested. Lack of comparison with recent inference-time or training-time hallucination mitigation approaches (e.g., DoLa [ICLR 2024], Integrative Decoding [Cheng et al., ICLR 2025], Mask-DPO [ICLR 2025]) weakens claims of superiority.

2. The framework increases inference-time cost substantially, but quantitative trade-offs (runtime vs. factual gain) are not reported. This omission limits the paper’s practical relevance for deployment.

**Questions:**

1. Can the authors provide quantitative runtime scaling (e.g., FLOPs or wall-clock ratios) relative to naïve decoding? This would clarify practical feasibility.

2. Why were approaches like DoLa, FactAlign, or Mask-DPO not included for comparison? Even small-scale or qualitative comparisons would strengthen the argument.

---

> ### Author Response · Authors · 2025-12-03
>
> > Lack of comparison with recent inference-time or training-time hallucination mitigation approaches
>
> We have expanded our evaluation to include the suggested approaches to the greatest extent possible. The results confirm the superiority of our method. For the detailed results and analysis, please refer to the *Shared Response* at the top of this web page.
>
> > quantitative trade-offs (runtime vs. factual gain) are not reported
>
> The trade-offs between factual gain and runtime is demonstrated in Figure 1 (b) in the initial manuscript. We kindly refer the reviewer to the *Shared Response* for the detailed efficiency analysis.
>
> > Can the authors provide quantitative runtime scaling (e.g., FLOPs or wall-clock ratios) relative to naïve decoding?
>
> The quantitative runtime scaling relative to naive decoding is reported in Figure 1 (b) in the initial manuscript. We kindly refer the reviewer to the *Shared Response* for the detailed efficiency analysis.
>
> > Why were approaches like DoLa, FactAlign, or Mask-DPO not included for comparison?
>
> We have expanded our evaluation to include the suggested approaches to the greatest extent possible. The results confirm the superiority of our method. For the detailed results and analysis, please refer to the *Shared Response* at the top of this web page.

---

### Author Response · Authors · 2025-12-03
**Shared Response**

We thank the reviewers and ACs for their time and constructive feedback. We present here several important new results and address key shared concerns. We have also revised the manuscript to incorporate the suggestions and additional findings. We acknowledge that the revised version exceeds the page limit and will ensure it adheres to the required limits during the production stage if accepted.

# SOTA Hallucination Mitigation Results

We have expanded our evaluation to include additional baselines, specifically the inference-time methods DoLa [1], Integrative Decoding [2], ChatProtect [3], and Self-Endorse [4], as well as the training-time method FactAlign [5]. In Table 1 in the revised manuscript, our method achieves the lowest number of unsupported claims and the highest factual precision among all inference-time hallucination mitigation approaches across multiple models. **It improves factual precision by approximately 25% over Greedy decoding and by around 15% over the second-best method**, Self-Endorse. In Table 3, **our method also provides strong complementary benefits to FactAlign, increasing factual precision by roughly 25% when the two methods are combined.**

# Efficient Scaling of Factual Precision

Our method introduces some computational overhead, but it remains significantly more efficient than competitive approaches such as ChatProtect and Self-Endorse. Across both plots in Figure 3 in the revised manuscript, **SHARS consistently achieves a superior factuality–efficiency trade-off**.

For Qwen3-32B, SHARS operates within a relatively low runtime range (approximately 10–40×) while attaining higher factual precision (around 65–78%). In contrast, Self-Endorse requires substantially higher runtime (35–60×) to reach similar or lower precision, and ChatProtect adds further computational cost with only modest gains over the Greedy baseline. The trend is even clearer for Llama3.1-8B: SHARS maintains strong precision (about 68–78%) at runtimes below 50×, whereas Self-Endorse exceeds 350× runtime to achieve comparable accuracy of 71%. Overall, the results demonstrate that SHARS delivers consistently higher precision at markedly lower computational cost, making it substantially more efficient than existing methods.

In addition, we expect the computational cost of our method to be substantially reduced by designing more efficient hallucination detectors, such as Linear Probes [6]. We leave this exploration to future work.


*References*:

1. Chuang, Yung-Sung, et al. "DoLa: Decoding by Contrasting Layers Improves Factuality in Large Language Models." The Twelfth International Conference on Learning Representations. 2023.
2. Cheng, Yi, et al. "Integrative Decoding: Improving Factuality via Implicit Self-consistency." The Thirteenth International Conference on Learning Representations. 2024.
3. Mündler, Niels, et al. "Self-contradictory Hallucinations of Large Language Models: Evaluation, Detection and Mitigation." The Twelfth International Conference on Learning Representations. 2023
4. Wang, Ante, et al. "Improving LLM Generations via Fine-Grained Self-Endorsement." Findings of the Association for Computational Linguistics ACL 2024. 2024.
5. Huang, Chao-Wei, and Yun-Nung Chen. "FactAlign: Long-form Factuality Alignment of Large Language Models." Findings of the Association for Computational Linguistics: EMNLP. 2024.
6. Obeso, Oscar, et al. "Real-time detection of hallucinated entities in long-form generation." arXiv preprint arXiv:2509.03531 (2025).

---

### Meta-Review · Area_Chair_Knfa · 2025-12-22

**Summary:**

This paper proposes SHARS (Step-wise Hallucination Rejection Sampling), an inference-time scaling framework for mitigating hallucination snowballing in long-form generation. The key idea is to detect and reject hallucinated content at intermediate steps during decoding, thereby preventing error propagation. The paper further instantiates this framework with HalluSE, an uncertainty-based hallucination detector tailored for long-form generation.

**Reviewer Concerns:**

Concerns partially addressed by the rebuttal:
- The lack of baseline comparisons and efficiency analysis was largely addressed by adding strong inference-time and training-time baselines and by providing clearer runtime–factuality trade-off analyses.
- Additional experimental results on a smaller model partially mitigate concerns about dependence on large instruction-following models.
- Concerns about positioning relative to the broader uncertainty and hallucination literature were largely addressed by clarifying that uncertainty estimation is not the primary contribution and by reframing the hallucination detector as a replaceable component.

Concerns still outstanding:
- While the rebuttal clarifies the use of an external NLI model, the appropriateness of the “training-free” and “no external models” framing remains disputed.
- Fundamental disagreement remains regarding the depth and nature of the contribution, with some reviewers viewing SHARS as an inference-time orchestration or prompt-driven pipeline rather than a substantial methodological advance.
- The framework’s reliance on multiple subcomponents and its generality and robustness remain unconvincing to these reviewers.

**Reviewer Scores:**

- **Reviewer frm6**: 6
  This reviewer’s main concerns were missing baseline comparisons and unclear efficiency trade-offs. These points were substantially addressed in the rebuttal and revised manuscript. I believe this reviewer would likely have increased their score to the 6.

- **Reviewer MR2W**: 6
  While initially concerned about inference-time cost and practicality, this reviewer acknowledged the empirical strength of the results. Given the additional efficiency analysis and expanded experiments, I expect this reviewer would also have moved to a 6.

- **Reviewer daAC**: 2
  This reviewer raised fundamental concerns about the depth of methodological contribution, lack of theoretical grounding, and characterization of the approach as largely prompt-based. These core objections do not appear to be resolved by the rebuttal. I do not expect a meaningful score change, and the reviewer would likely remain at 2.

- **Reviewer y9bF**: 2
  This reviewer questioned the methodological novelty, reliance on auxiliary components (e.g., NLI models), and venue fit relative to ICLR standards. Although some clarifications were provided, the rebuttal does not seem sufficient to change their overall assessment. I expect this reviewer would also remain at 2.

---

### Decision · Program_Chairs · 2026-01-26

Reject